# Impact of Weight Carriage on Joint Kinematics in Asian Elephants Used for Riding

**DOI:** 10.3390/ani11082423

**Published:** 2021-08-17

**Authors:** Siriphan Kongsawasdi, Janine L. Brown, Khajohnpat Boonprasert, Pornsawan Pongsopawijit, Kittichai Wantanajittikul, Siripat Khammesri, Tanapong Tajarernmuang, Nipaporn Thonglorm, Rungtiwa Kanta-In, Chatchote Thitaram

**Affiliations:** 1Department of Physical Therapy, Faculty of Associated Medical Sciences, Chiang Mai University, Chiang Mai 50200, Thailand; siriphan.k@cmu.ac.th (S.K.); tanapong.tajarernmuang@gmail.com (T.T.); n.thong19@gmail.com (N.T.); rungtiwa.tik@gmail.com (R.K.-I.); 2Center of Elephant and Wildlife Health and Research, Chiang Mai University, Chiang Mai 50200, Thailand; BrownJan@si.edu (J.L.B.); khajohnpat@gmail.com (K.B.); pornsawan58@gmail.com (P.P.); aniloverspat@gmail.com (S.K.); 3Center for Species Survival, Smithsonian Conservation Biology Institute, Front Royal, VA 22630, USA; 4Department of Companion Animals and Wildlife Clinics, Faculty of Veterinary Medicine, Chiang Mai University, Chiang Mai 50100, Thailand; 5Department of Radiologic Technology, Faculty of Associated Medical Sciences, Chiang Mai University, Chiang Mai 50200, Thailand; kittichai.wan@cmu.ac.th

**Keywords:** elephant, tourism, riding, weight carriage, gait, saddle, kinematics, gait

## Abstract

**Simple Summary:**

Riding elephants is one of the most controversial activities in the tourist industry, with concerns over whether load carrying is physically harmful. Here, we used an empirical approach to test how carrying loads up to 15% of the elephant’s body mass affected gait parameters. The maximal angles of fore- and hindlimb joints of elephants walking at a normal, self-selected speed carrying a mahout only were first evaluated and then compared to those walking with a saddle carrying two people plus added weight to reach a 15% body mass load. Data were analyzed using a computerized three-dimensional inertial measurement system. There were no significant differences between movement angles, including flexion, extension, abduction, and adduction of the fore- or hindlimbs between these two riding conditions. Thus, we found no evidence that carrying two people in a saddle causes significant changes in gait patterns or potentially affects musculoskeletal function. More studies are needed to further test longer durations of riding on different types of terrain to develop appropriate working guidelines for captive elephants. Nevertheless, elephants appear capable of carrying significant amounts of weight on the back without showing signs of physical distress.

**Abstract:**

Background: Elephants in Thailand have changed their roles from working in the logging industry to tourism over the past two decades. In 2020, there were approximately 2700 captive elephants participating in activities such as riding and trekking. During work hours, riding elephants carry one or two people in a saddle on the back with a mahout on the neck several hours a day and over varying terrain. A concern is that this form of riding can cause serious injuries to the musculoskeletal system, although to date there have been no empirical studies to determine the influence of weight carriage on kinematics in elephants. Methods: Eight Asian elephants from a camp in Chiang Mai Province, Thailand, aged between 21 and 41 years with a mean body mass of 3265 ± 140.2 kg, were evaluated under two conditions: walking at a normal speed without a saddle and with a 15% body mass load (saddle and two persons plus additional weights). Gait kinematics, including the maximal angles of fore- and hindlimb joints, were determined using a novel three-dimensional inertial measurement system with wireless sensors. Results: There were no statistical differences between movement angles and a range of motion of the fore- and hindlimbs, when an additional 15% of body mass was added. Conclusion: There is no evidence that carrying a 15% body mass load causes significant changes in elephant gait patterns. Thus, carrying two people in a saddle may have minimal effects on musculoskeletal function. More studies are needed to further test longer durations of riding on different types of terrain to develop appropriate working guidelines for captive elephants. Nevertheless, elephants appear capable of carrying significant amounts of weight on the back without showing signs of physical distress.

## 1. Introduction

Elephants in Thailand (Elephas maximus) under human care have changed from working primarily in the logging industry to being used in tourism, since logging was banned in 1989. There are more than 3500 elephants in Thailand; 95% are privately owned, and of those, 77% are kept in tourist camps (National Elephant Institute, Thailand). Elephants work in a variety of activities, such as riding, shows, bathing, and trekking [1]. Elephant riding involves one to two people sitting in a howdah (i.e., saddle) or directly on the back or neck (i.e., bareback) [2]. A large study in Northern Thailand found that during work hours, elephants carry tourists for 2–3 h and walk on average 4–5 km per day over varying types of terrain [1]. Of all the activities that tourist elephants participate in, none has received more criticism in the popular press than riding, with intimations that it causes long-term damage to their physical health. In Thailand, elephant welfare management falls under the Prevention of Cruelty and Animal Welfare Provision Act B.E.2557 (A.D. 2014), which states that weight bearing for riding should not exceed 10% of the elephant’s body mass or 350 kg [3]. However, to our knowledge, there have been no scientific studies on the consequences of different levels of weight bearing on elephants and whether carrying tourists affects their musculoskeletal health or gait function.

The effects of load carriage on the back have been studied extensively in humans [4,5,6,7,8] and horses [9,10], with findings showing heavy loads can cause structural changes in the spine leading to spinal disorders [4]. In a standing posture, heavy loads exert downward pressures on the spine, hip, knee, and ankle, causing an abnormal posture, i.e., increased forward lean of trunk and lateral pelvic tilt to maintain the center of mass [5,9]. Alterations in kinematic patterns may help reduce joint stress by absorbing the impact of forces on joints, with adjustments aimed to maintain balance during walking with additional loads [4,5]. Such biomechanical alterations, however, can cause damage to cartilage and soft tissue degeneration and in time, lead to ligament injuries, osteoarthritis, and pain [5,7,11]. Additionally, vertical and anteroposterior ground reaction forces (GRFs) have been shown to increase proportionally to backpack loads and are a major risk factor for overuse injuries [12].

Kinematic gait measurements have been used widely by clinicians and researchers to qualitatively assess the degree of movement impairments resulting from musculoskeletal and neurological problems. Earlier kinematic studies of gaits in horses, dogs, and elephants [13,14,15,16] utilized two-dimensional motion analysis with video capture. Recently, a novel wireless inertial measurement unit (IMU) has been developed with a motion sensor that enables three-dimensional (3D) analyses of a body’s specific force, movement, and angular rate and is now widely used in studies of motion analysis [17,18]. Linear acceleration is determined using accelerometers, rotational velocities are evaluated using gyroscopes, and a magnetometer is used as a heading reference. Due to their small size and being wireless, IMUs have the potential for dynamic movement analyses in the field as well as in the laboratory. In 2004, Keegan et al. [18] compared a sensor-based accelerometer and gyroscope system to a video-based motion analysis for detecting equine lameness and found a high correlation between the two methods (r2 = 0.82–0.95) with good to excellent agreement (k = 0.56–0.76) for fore- and hindlimbs, respectively, although the inertial sensor system performed better for the latter. Later, McCracken et al. [19] reported an inertial sensor system utilizing two accelerometers at the head and the pelvis, and a uni-axial gyroscopic sensor at the forelimb was better at detecting lameness in horses compared to subjective analyses by veterinarians.

Compared to horses and humans, there have been relatively few biomechanical studies of elephants. Most have explored normal kinematic and kinetic parameters [13,15,20], revealing that elephants walk and maintain stability using lateral sequence footfall and inverted pendulum patterns to economize muscular work and energy expenditure. Ren and Hutchinson, who developed an IMU system, used it to quantify locomotor patterns at normal and faster speeds in Asian and African elephants, measuring the center of mass motion and torso rotation. They revealed that kinematic and kinetic patterns were similar and that the metabolic cost of walking was minimal at a normal speed, but it changed gradually with the increasing speed of locomotion from inverted pendulum at slow speeds to a more bouncing gait at increasing speeds. Elephants retained a lateral sequence footfall pattern, which helped modulate energy output [20,21]. Modified gait behavior has been shown to be associated with musculoskeletal pathologies in elephants, so gait assessments could help detect subtle problems associated with lameness [18]. To our knowledge, no studies have examined the effect of load carriage on biomechanical changes of locomotion in elephants. Therefore, this study was designed to measure how biomechanics differed in tourist elephants walking with only a mahout (i.e., handler) on the neck compared to those with an additional weight load of 15% of the elephant’s body mass.

## 2. Materials and Methods

### 2.1. Animal Ethical Consent

This study was approved by The Animal Care and Use Committee of the Faculty of Veterinary Medicine (FVM-ACUC), Chiang Mai University (Research ID R5/2018; 11 July 2018).

### 2.2. Study Animals

Eight healthy adult elephants (6 females and 2 males) with an average age of 33.4 ± 10.2 years (range: 20–51 years) from a private elephant camp in Chiang Mai Province were included in the study (Table 1). The average body mass was 3265 ± 140.2 kg (range: 3112–3450 kg) measured with all four limbs on a truck scale (±5 kg) (Matratham Scales Ltd., Bangkok, Thailand). The elephants routinely worked as tourist trekking animals with a saddle on the back carrying one or two people between 8:00 and 15:00 for no more than 5 h per day, with breaks in between riding rounds. During the day, when not working, the elephants were tethered on a 3 m chain in a shed with other females in a group and fed primarily napier grass (Pennisetum purpureum), bana grass (Pennisetum purpureum X, P. Americanum hybrid), corn stalks, and water several times a day. Food supplements consisted of banana, sugar cane, hay, and rice. The animals were given an annual physical examination by the staff veterinarian. The subjects were further deemed clinically healthy by experienced veterinarians from the Center of Elephant and Wildlife Health and Research, Faculty of Veterinary Medicine, Chiang Mai University; i.e., they showed normal appetites and gait patterns, no apparent skin lesions, and no outward joint problems. The experimental protocol stated that a trial would be cancelled if an elephant showed signs of fatigue and lameness or did not walk; none were excluded from this study.

### 2.3. Experimental Design

Data on gait parameters were collected when walking normally with only a mahout on the neck (approximately 50–70 kg) followed by an addition of a 15% body mass load, which entailed carrying two people in a wooden saddle plus additional free weights as needed. The elephants were rested at least 10 min between trials to avoid fatigue. The elephants walked at a steady speed in a straight line for 40 m on a flat grass field at their normal pace. To reduce the effect of an acceleration phase at the beginning and a deceleration phase at the end of each trial, data were collected at the midpoint of the walking path, when the gait speed was more consistent (20 m). A video camera (Logitech C925-e Webcam, Lausanne, Switzerland; resolution: 640 pixels × 480 pixels; refresh frame rate: 30 Hz) also captured walking activity (Figure 1a).

### 2.4. Data Collection and Analysis

Kinematic measurements of the gait cycle were composed of two phases: stance phase—when the foot contacted the ground and provided limb support; and swing phase—period of time when the foot was off the ground and moving forward. Data collected included measures of maximal angles of the sagittal plane; flexion extension; frontal plane (side to side movement); abduction−adduction and transverse plane; the rotation of shoulders and hips; and the flexion extension of the elbow and knee.

Data were captured using a 3D IMU system (STT Ingeniería Y Sistemas, San Sebastián, Spain), which consisted of iSen™ 3.01 software and STT-IWS WiFi inertial sensors. Joint angles were obtained from the 3D coordinate integration of the acceleration, the angular rate and the magnetic field vector by an algorithm in the IMU software, which used a rotation matrix based on a local axis with a common global system reference. Each body segment had a local reference frame, and the rotational axis was aligned to the local axis of this reference. Then, the angle was calculated from two vectors from the local reference frame. An accelerometer served as the primary sensor responsible for measuring inertial acceleration over time. A gyroscope acted as an inertial sensor that measured the elephant’s angular rate with respect to an inertial reference frame. The magnetometer sensor measured the strength and direction of the magnetic field. The IMU software also measured and reported specific gravity. Each elephant was equipped with five IMU sensors secured with strips of a surgical tape: two at the forelimb along the proximal humerus and the radius bone, two at the hindlimb along the proximal part of femur and tibia, and one reference sensor placed at the caudal border of scapula (Figure 1b). The sensor signals were transmitted to the software via the wireless technology. One person was responsible for sensor placement on all elephants to reduce variation. Methodology was similar to the IMU method developed by Ren and Hutchinson [21].

### 2.5. Statistical Analysis

Descriptive data were expressed as the median and standard error (SE). The Wilcoxon signed-rank test, a nonparametric statistical test, was used to determine gait characteristic differences between the two load-bearing conditions. Statistical analyses were performed using SPSS Statistics (version 26).

## 3. Results

The median walking speeds of elephants with only a mahout on the neck (no load) and with an additional 15% of body mass load were 0.96 ± 0.7 and 0.73 ± 0.14 m/s, respectively, and did not differ (*p* = 0.068).

### Angle of Movement during a Gait Cycle

Kinematic analysis determined there were no significant changes in the angles of the fore- or hindlimb joints when walking with a 15% additional body mass load (p > 0.05) (Table 2). Similar to the flexion and extension angles, the other movements of the fore- and hindlimbs showed no differences when compared between the unloaded and 15% additional weight-loaded walking conditions (*p* < 0.05) (Table 3). The data of subjects’ movement angles (individual and group) are given as Appendix A in Appendix A respectively.

## 4. Discussion

Our study is the first to evaluate the influence of weight carriage on riding elephants using a novel wireless 3D IMU sensor system to assess joint angles during walking cycles at a normal speed. The results showed that neither walking speeds nor movement angles (i.e., flexion, extension, abduction, and adduction) of the fore- and hindlimbs during stance or swing phases were affected by loads up to 15% of the elephant’s body mass. We chose a higher loading percentage than the recommended 10% [3] because not all camps abide by that recommendation, so some elephants could inadvertently be overloaded to that degree during trekking. As a result, we wanted to err on the side of a maximal load, which in this case showed no undo effects of even that amount of weight. These findings agree with studies in horses where stride parameters such as the symmetry and range of limb movements were not affected by carrying extra loads. For example, carrying weights up to 30% of body mass in Arabian horses [22], 29% in Japanese native horses [23], and 25% in light horses [24,25] resulted in no noticeable effects on behavior or physiological parameters, such as heart rates and plasma lactate concentrations. Therefore, it could be implied that for elephants, carrying 15% of body mass loads does not impact musculoskeletal function, at least during short walking bouts on flat terrain.

Because quadruped animals such as the horse rely on the anatomical coupling between front and hind limbs, carrying loads can in fact lead to changes in joint angles and range of motion (ROM) under certain conditions [9,26]. Physiological and biomechanical gait changes have been associated with a rider and added saddle weight [9,27]. At the walk and the trot, there was a significant influence on back kinematics in horses with a saddle plus 75 kg compared to those with a lunging girth or saddle only [9]. Limb kinematics also were altered, with an observed increase in forelimb retraction to provide more support for the body. Gunnarson et al. [10] investigated the effects of rider weight plus an additional 20–35% of the horse’s body mass on gait parameters during a trot and found that stride duration was shortened with increasing weight and unipedal support in the hindlimbs was higher than that in the forelimbs. There also was a positive linear relationship between weight ratio and stride frequency and a negative linear relationship between weight ratio and stride length. Another kinetic study of trotting horses found an increase in the absolute peak vertical GRF of fore- and hindlimbs when carrying a rider as compared to in-hand [28]. Back loading has been shown to increase back extension in horses, which may contribute to soft tissue injuries; however, that was only observed at the canter, and even then, there were no changes in limb kinematics [9]. Elephants giving rides go at a self-directed pace and are not pushed by the mahout to walk faster or to “trot”, so perhaps, it should be not surprising that gait kinematics were unaltered by increased back loading in the elephants of this study.

In humans, carrying heavy loads on the back can cause structural changes and alter gait kinematics. For example, heavy backpacks carried by military personnel, recreational hikers, and students can create stress and tension in back muscles and ligaments that result in lower limb kinematic gait changes [5,8,29]. As loads increase, walking velocity and cadence (the number of steps per minute) generally decrease, while stride length (the distance between successive ground contacts of the same foot) and step length (the distance between the point of initial contact of one foot and the point of initial contact of the opposite foot) shorten, and the period of double support (when both feet are in contact with the ground to provide greater stability) increases as a consequence of decreasing stride length to provide greater stability [5,6,7]. Carrying loads from 20–60% of body mass resulted in participants walking at progressively slower speeds with decreased stride lengths [6,29,30]. A systematic review and meta-analysis of 54 studies revealed significant impacts of back carriage that included increasing hip, knee, and ankle ROM resulting from increased speeds of movement and stride length [31]. Hip and knee ROM increased with a higher knee lift, resulting in increased hip and knee flexion at heel strike. Raising hips and knees higher may be a means of absorbing impact forces associated with additional loads [5,29]. While there could be weight-bearing differences between bipedal and quadrupedal mechanisms, most species (bipedal, quadrupedal, and even hoppers, such as kangaroos), including humans, use an inverted pendulum mechanism to modulate the center of mass during walking. Fore- and hindquarters move in a coordination interlimb pattern and are synchronized by vaulting over the respective stance limbs, minimizing muscular work and the metabolic costs of locomotion [32,33]. Kinematics gait studies on elephants have revealed they use a similar lateral sequence footfall pattern to conserve energy and limit muscular activity [15,34,35]. Ren et al. [20] used an IMU-based method to quantify kinematic and kinetic mechanisms and showed elephants use an effective energy-saving locomotion style by changing the center of the mass pattern. The limbs’ effective mechanical advantage is increased by straightening the limbs, thus reducing GRF moment arms. Our earlier work showed the distance of each stride for both fore- and hindlimbs was similar at 200 cm each at a comfortable walking speed [13]. Therefore, even with a 15% added load, elephants appeared capable of maintaining a lateral sequence footfall pattern and inverted pendulum mechanism at a normal speed, distributing the center of mass proportionally across all four limbs, thereby conserving energy and adding to stability.

## 5. Conclusions

During load carriage on the back, changes in posture, limb kinematics and gait often result and can lead to increased metabolic demand, muscle fatigue, and in the long-term, muscle and joint disorders. Our results suggest that loading an elephant’s back to the degree used in this study does not significantly affect gait characteristics over a short distance and on flat terrain. Assumably, they retain a normal lateral sequence and a footfall pattern to maintain stability during the transitions of movement [15,34,35]. Therefore, riding elephants may have minimal effects on musculoskeletal function if weight, riding duration, floor substrate and slope, and saddle padding are appropriate. Riding as a form of exercise has been associated with better body condition, metabolic health [36], and lower fecal glucocorticoid metabolites concentrations in tourist elephants [37]. In this study, an additional 15% of body mass load did not alter kinematics; thus, recommendations by the Minister of Agriculture and Cooperatives that elephant weight bearing should not exceed 10% may be reasonable from a physiological standpoint [3].

There were a few limitations of the study. Measurements were taken only on one side of the elephant (left), and specific changes to the vertebrae, pelvis, and carpal and tarsal joints were not evaluated, as often is done in horses [18,38]. We also did not have a reference sensor at the pelvis to more accurately measure hindlimb angles. Elephants were encouraged to walk by a mahout riding on the neck, which was considered “no load” because the mahout’s weight was small (~60 kg) and they were not positioned directly on the back as were the saddles. A truly self-directed walking pattern captured by closed circuit television (CCTV), however, might show a more natural gait pattern, which is worth further investigation. Another limitation is that kinetics might be more important for elephants than kinematics, e.g., forces and moments. Kinematics may not change, but kinetics could vary as the masses do (F = m × a). Perhaps, the IMU data could address this, as other studies of locomotion have done. It should be noted that these elephants have worked as trekking elephants for years, so additional studies on elephants that have never carried heavy loads are needed to determine if these animals have learned to adjust their gaits to minimize strain and energetic demands. Further investigations are needed using larger numbers of elephants to explore longer-term impacts of saddle riding on elephant structural and locomotor health, for instance when walking over varying terrain and longer distances as is typically the case in Thailand. Nevertheless, there is no evidence that carrying a 15% body mass load causes significant changes in elephant gait patterns, so carrying two people in a saddle may have minimal effects on musculoskeletal function.

## Figures and Tables

**Figure 1 animals-11-02423-f001:**
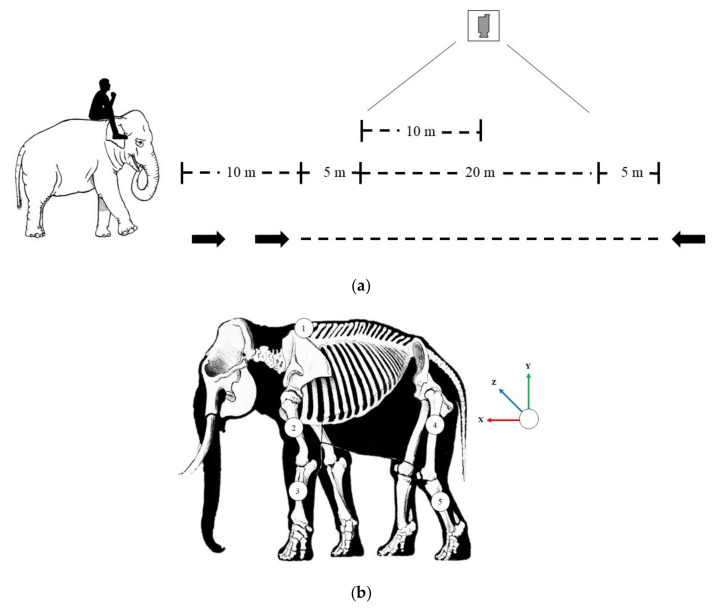
(**a**) Schematic of how gait biomechanics data were collected and positions of the five inertial measurement units at the fore- and hindlimbs. (**b**) A trackway-aligned system determined by arranging the X, Y, and Z axes with the trackway.

**Table 1 animals-11-02423-t001:** Demographic data and weight carriage loads of eight Asian elephants used in this study.

ID.	Age (yr)*)*	Sex	Height (cm)	Body Mass (kg)	Weight Added (kg) ^1^
EM1	21	Female	280	3425	523
EM2	36	Female	240	3450	525
EM3	51	Female	232	3200	493
EM4	31	Female	250	3280	493
EM5	31	Female	235	3146	472
EM6	20	Male	270	3120	468
EM7	36	Male	260	3386	507
EM8	41	Female	230	3112	467
Mean	33.4		249.6	3264.9	493.5
SD	10.2		18.7	140.2	23.5

^1^ Weight load equal to 15% of each animal’s body mass.

**Table 2 animals-11-02423-t002:** Median (±standard error (SE)) maximal flexion and extension angles of fore- and hindlimb joints and changes in Asian elephants walking with only a mahout on the neck (no load) and with an additional 15% of body mass load.

Angle	Forelimb (Degree)				Hindlimb (Degree)		
	No Load	15% Load	*p*-Value	No Load		15% Load	*p*-Value
Proximal Flexion	16.71 ± 3.05	20.13 ± 2.42	0.401	16.65 ± 3.28		17.00 ± 2.38	0.575
Proximal Extension	23.88 ± 4.44	21.90 ± 3.37	0.779	21.87 ± 3.47		22.32 ± 3.40	0.889
Distal Flexion	14.90 ± 2.35	19.47 ± 2.94	0.327	11.58 ± 1.93		16.73 ± 1.44	0.484
Distal Extension	20.63 ±9.74	12.57 ± 2.82	0.161	15.49 ± 2.00		12.72 ± 1.67	0.674

No difference at *p* < 0.05; Wilcoxon signed-rank test.

**Table 3 animals-11-02423-t003:** Median (±SE) proximal angles and rotations in Asian elephants walking with only a mahout on the neck (no load) and with an additional 15% of body mass load.

Angle	Forelimb (Degree)			Hindlimb (Degree)		
	No Load	15% Load	*p*-Value	No Load	15% Load	*p*-Value
Adduction	22.10 ± 3.24	23.54 ± 3.65	0.401	24.53 ± 1.96	24.97 ± 3.64	0.779
Abduction	19.47 ± 2.14	16.24 ± 3.28	0.575	17.97 ± 2.27	18.28 ± 3.39	0.674
Internal rotation	18.81 ± 3.71	20.18 ± 3.42	0.674	15.83 ± 3.74	17.73 ± 2.54	0.484
External rotation	18.54 ± 2.18	19.59 ± 1.99	0.327	16.93 ± 2.91	15.83 ± 3.87	0.484

No difference at *p* < 0.05; Wilcoxon signed-rank test.

## Data Availability

The data presented in this study are available on request from the first author.

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
