# Peer review of "Impact of Weight Carriage on Joint Kinematics in Asian Elephants Used for Riding"

_animals, 2021, doi:10.3390/ani11082423_

Round 1

Reviewer 1 Report

Revisions are a good improvement but not all suggestions made it into the MS itself, just being argued in the Response- the points below should be clarified in the paper in revision:

(new comment) If video footage was available why was calculating speed not feasible? No scale object in view of the video? The IMU system cannot calculate median speed from accelerations?

The IMU methods are not explained sufficiently; it is being treated as a black box. More effort needs to be invested into obtaining and explaining these methods. My original comment has not adequately been addressed in the MS-- "Please explain in more detail, ideally with a figure, how the angles were computed. A 3D coordinate system showing the world, body and segment CSs should be made explicit. Better explanation for readers
that do not know these IMUs is needed to explain how one goes from the raw data to the angles. As written, it is not possible to precisely determine what each angle is relative to in the global/local CSs (e.g. in an elephant's anatomical context)."

--gravity is explained in the Response as the IMUs' angle's reference but this seems not explained in the MS-- please do so as part of the Methods detail requested above.

Referring to angles as proximal and distal only adds to the confusion, as now there is little anatomical context.

My other point was not adequately addressed in the MS- "A limitation not noted is that kinetics might be more important for elephants than kinematics; e.g. forces and moments. Kinematics may not change but surely the kinetics do as the masses do (F= m*a). Perhaps even IMU data could address this; other studies of locomotion have done so."

The authors do not seem to understand that as because their IMUs give acceleration data, those could be used (if experimental design suited it) to estimate kinetics, via (e.g., F= m * a) basic dynamics. It is not correct to say that kinematics is "specific and more appropriate in field work"-- it just happens that this is all that the study has reported, but different applications of IMUs and data like those in this study could address kinetics. This would be easy to at least acknowledge in the MS Discussion; and the shared data from the study might even allow someone to do such analyses with the current data, as it's not impossible to do. (and thank you for sharing the data-- that's good science!)

Author Response

Reviewer#1

Revisions are a good improvement but not all suggestions made it into the MS itself, just being argued in the Response- the points below should be clarified in the paper in revision:

(new comment) If video footage was available why was calculating speed not feasible? No scale object in view of the video? The IMU system cannot calculate median speed from accelerations?

Answer: We found the videos and calculated walking speeds using that as follows:

Page 5 Line 152-153.

Median walking speed of elephants with only a mahout on the neck (no load) and with an additional 15% of body mass load was 0.96±0.7 and 0.73±0.14 m/sec, respectively, and did not differ (p=0.068).

The IMU methods are not explained sufficiently; it is being treated as a black box. More effort needs to be invested into obtaining and explaining these methods. My original comment has not adequately been addressed in the MS-- "Please explain in more detail, ideally with a figure, how the angles were computed. A 3D coordinate system showing the world, body and segment CSs should be made explicit. Better explanation for readers that do not know these IMUs is needed to explain how one goes from the raw data to the angles. As written, it is not possible to precisely determine what each angle is relative to in the global/local CSs (e.g. in an elephant's anatomical context)."

--gravity is explained in the Response as the IMUs' angle's reference but this seems not explained in the MS-- please do so as part of the Methods detail requested above.

Answer: Thank you for your comment and suggestion. We have tried to add more explanation of the IMU method and have added several sentences, as indicated below. We acknowledge that we are not engineers and had to rely on manufacturer information, which unfortunately was not highly detailed. So, yes, the reviewer is right that the IMU was a black box in our hands and that we followed manufacturer protocols to conduct our study. Our requests for more detailed information from the company on the unit itself and the algorithm it used have yet to be answered, so we have added the information we could find. We hope this will be enough for the reviewer.

Page 4 Line 127-134

Joint angles were obtained from three-dimensional coordinate integration of acceleration, angular rate and the magnetic field vector by an algorithm in the IMU software, which used a rotation matrix based on a local axis with a common global system reference. Each body segment had a local reference frame and the rotational axis was aligned to the local axis of this reference, then the angle was calculated from two vectors from the local reference frame. An accelerometer served as the primary sensor responsible for measuring inertial acceleration over time. A gyroscope acted as an inertial sensor that measured the elephant's angular rate with respect to an inertial reference frame. The magnetometer sensor measured the strength and direction of a magnetic field. The IMU software also measured and reported specific gravity.

Referring to angles as proximal and distal only adds to the confusion, as now there is little anatomical context.

Answer: We have added XYZ axes to Figure 1 and included that in the caption as follows:

Figure 1. positions of the five inertial measurement units at the fore- and hindlimbs. A trackway aligned system was determined by arranging the XYZ axes with the trackway (b).

My other point was not adequately addressed in the MS- "A limitation not noted is that kinetics might be more important for elephants than kinematics; e.g. forces and moments. Kinematics may not change but surely the kinetics do as the masses do (F= m*a). Perhaps even IMU data could address this; other studies of locomotion have done so."

Answer:  We agree and have added the limitation statement as recommended.

Page 7 Lines 244-246

Another limitation is that kinetics might be more important for elephants than kinematics; e.g. forces and moments. Kinematics may not change but kinetics could as the masses do (F= m*a). Perhaps the IMU data could address this, as other studies of locomotion have done.

The authors do not seem to understand that as because their IMUs give acceleration data, those could be used (if experimental design suited it) to estimate kinetics, via (e.g., F= m * a) basic dynamics. It is not correct to say that kinematics is "specific and more appropriate in field work"-- it just happens that this is all that the study has reported, but different applications of IMUs and data like those in this study could address kinetics. This would be easy to at least acknowledge in the MS Discussion; and the shared data from the study might even allow someone to do such analyses with the current data, as it's not impossible to do. (and thank you for sharing the data-- that's good science!)

Answer: Thank you for your comment and suggestion. We have removed the comment that kinematics is ‘specific and more appropriate in field work’. Instead, we say “Due to their small size and being wireless, IMUs have the potential for dynamic movement analyses in the field as well as in the laboratory”.

Our team consisted of general veterinary practitioners and biologists, with no software engineer experience. Thus, we do not have the expertise to transform data algorithm from kinematics to kinetics. Our original idea was to compare the gait pattern of elephants with only a mahout on the neck (no load) and with an additional 15% of body mass load. But we are glad the reviewer appreciates our sharing the data, and hope others might be able to do other things with it.

Reviewer 2 Report

the major comment on this is the same in that there is no measure of speed of locomotion - speed affects gait in almost all animals so to not be able to control the speed makes the results unable to be compared across elephants - it seems from figure 1A that speed should be obtainable from the data the authors have, if you know the distance being travelled and the camera has a time on it (which virtually every camera does even if not synced to real time) then with the frame rate you can get speed at the very least you could get an approximate speed and use this to compare across the elephants. you can also use a known scale bar so even the size of the elephant head could be used as a scale bar in the video to get speed. 

im also curious as to why an experiment that set out to determine if human load carrying affected gait chose such high weights to add to the elephants 15% is around 450kg+ which is way over what the addition of 2 humans would be - this could be clarified in the paper 

Author Response

Reviewer#2

the major comment on this is the same in that there is no measure of speed of locomotion - speed affects gait in almost all animals so to not be able to control the speed makes the results unable to be compared across elephants - it seems from figure 1A that speed should be obtainable from the data the authors have, if you know the distance being travelled and the camera has a time on it (which virtually every camera does even if not synced to real time) then with the frame rate you can get speed at the very least you could get an approximate speed and use this to compare across the elephants. you can also use a known scale bar so even the size of the elephant head could be used as a scale bar in the video to get speed. 

Answer: Thank you for your comment and suggestion. The VDO and data were obtained and a speed was calculated from that. This was acknowledged by adding the sentence in the Results.

Page 5 line 152-153

Median walking speed of elephants with only a mahout on the neck (no load) and with an additional 15% of body mass load was 0.96±0.7 and 0.73±0.14 m/sec, respectively, and did not differ (p=0.068).

I am also curious as to why an experiment that set out to determine if human load carrying affected gait chose such high weights to add to the elephants 15% is around 450kg+ which is way over what the addition of 2 humans would be - this could be clarified in the paper.

Answer: Thank you for your comment and we agree that we used a higher weight load than normally recommended. However, we have personally seen elephants loaded with more than two people, not often in Thailand but in Myanmar and Nepal, where howdas used for packing carry up to 4 people plus supplies, so we know this kind of overloading can occur. Thus, we decided to err on the side of testing more weight than most elephants would experience in tourism, and then if we saw changes, we would know to design more experiments to determine at what percentage the changes are occurring. In this case, even 15% had no effect in our experimental design.

Page 5 line 179-182

We chose a higher loading percentage than the recommended 10% [3] because not all camps abide by that recommendation and so some elephants could inadvertently be overloaded to that degree during trekking. As a result, we wanted to err on the side of a maximal load, which in this case showed no undo effects of even that amount of weight.

This manuscript is a resubmission of an earlier submission. The following is a list of the peer review reports and author responses from that submission.

Round 1

Reviewer 1 Report

Investigation into this area is necessary and important and the need for studies such as this is huge, given the numbers of elephants involved in riding camps in Thailand, and assumingly wider in Asia. the potential for application of this study is high. 

However there is a fundamental concern of the study design which must not be overlooked. the authors state the are evaluating the influence of weight carriage on gait kinematics. the elephants used in this study had been involved in elephant riding activities for a number of years and therefore the long term effects of weight carrying cannot be evaluated. it may well be that gate change had already taken place as a result of years of weight carrying. Secondly, the comparative study subject still was carrying a weight (a handler on the neck), this may well alter movement. Finally, the elephants are being examined walking in a very specific manor and thus minor alterations to locomotion may well be overlooked. 

Given the sensitivities of a study such as this, it would be valuable to consider these data as part of  a wider data set and compare perhaps via the use of CCTV to elephants walking at liberty, and elephants that were kept in an environment with considerably more free choice e.g. not chained overnight. 

Additionally, the number of animals in the study must be increased to strengthen the conclusions. 

Author Response

Reviewer#1

Comments and Suggestions for Authors

  1. Investigation into this area is necessary and important and the need for studies such as this is huge, given the numbers of elephants involved in riding camps in Thailand, and assumingly wider in Asia. the potential for application of this study is high. 

Answer: Thank you very much for recognizing the importance of this study.

  1. However there is a fundamental concern of the study design which must not be overlooked. the authors state they are evaluating the influence of weight carriage on gait kinematics. the elephants used in this study had been involved in elephant riding activities for a number of years and therefore the long term effects of weight carrying cannot be evaluated. it may well be that gate change had already taken place as a result of years of weight carrying.

Answer: We appreciate your comment and recognize that long-time use of elephants in riding might have had an effect already, although that has not been proven either. However, the goal of this study was to determine if carrying heavy loads has a direct effect on gait, which it did not. The arguments from animal rights groups has always been that riding (even after many years) is harmful and so this study was important from that standpoint.

To address this important comment, we have added a sentence to the end of the Conclusion section discussing study limitations that stated (Line 224-226)

“It should be noted that these elephants have worked as trekking elephants for years, so additional studies on elephants that have never carried loads are needed to determine if these animals have learned to adjust their gaits to minimize strain and energetic demands.”.

  1. Secondly, the comparative study subject still was carrying a weight (a handler on the neck), this may well alter movement.

Answer: Yes, we agree that elephants were still carrying a load in the form of its mahout. However, the body weight of the mahouts is small percentage compared to those of elephant (50-70 kg to 2500-3000 kg), and thus less than 2% of the elephant’s body weight. The mahout also sat behind the ears and not directly on the back, which differs from the orientation of the saddle, which was what we were most interested in. The paper by Ren et al. (2010) mentions gait analyses of Asian elephants with a rider on the neck. Moreover, our elephants were familiar with having a mahout on the neck, so we expected the change of gait pattern due to a mahout to have little or no effect. From a logistical standpoint, it would be difficult to have the elephant walk at a constant pace over the course we established in line with the camera and recording device. However, we agree another way to study this would be for the mahout to walk (run?) alongside the elephant, but that was not possible in this study.

Therefore. We add the sentence to the Conclusions section (Line 219-223)

“Elephants were encouraged to walk by a mahout riding on the neck [see 20], which was considered ‘no load’ because the mahout’s weight is small (~60 kg) and they are not positioned directly on the back as are the saddles. A truly self-directed walking pattern captured by closed circuit television (CCTV), however, might show a more natural gait pattern, and so is worth further investigation.”

  1. Finally, the elephants are being examined walking in a very specific manor and thus minor alterations to locomotion may well be overlooked. 

Answer: We agree and so have added this sentence to the Conclusions section, (Line 226-229)

“Further investigations are needed using larger numbers of elephants to explore longer-term impacts of saddle-riding on elephant structural and locomotor health; for instance, when walking over varying terrains and longer distances as is typically the case in Thailand.”.

  1. Given the sensitivities of a study such as this, it would be valuable to consider these data as part of a wider data set and compare perhaps via the use of CCTV to elephants walking at liberty, and elephants that were kept in an environment with considerably more free choice e.g. not chained overnight. 

Answer: We agree. See above answer to question 3. (Line 219-223)

  1. Additionally, the number of animals in the study must be increased to strengthen the conclusions. 

Answer: We had eight elephants in our study, which was low compared to domestic animals studies, such as horse, dog, cat, cattle. But this number was in line with other studies in elephants; e.g. Ren and Hutchinson 2008a (5 elephants); Ren et al. 2008, (15 elephants); Ren et al, 2010 (6 elephants); Hutchinson et al., 2006 (18 elephants).

We have added this sentence to the Conclusions, (Line 226-229)

 “Further investigations are needed using larger numbers of elephants to explore longer-term impacts of saddle-riding on elephant structural and locomotor health; for instance, when walking over varying terrains and longer distances as is typically the case in Thailand.”.

Reviewer 2 Report

the authors don't include any data on the speed the elephants were walking at other than 'normal' being used to classify it - if they didn't measure the speed then this is a serious flaw without speed information we can't assess the data in terms of the influence of the loads changes in speed will change gait so unless the speeds are the same with and without loads you can't answer the question posed 

if these data can be added then the manuscript can be re written 

Author Response

Reviewer#2

Comments and Suggestions for Authors

  1. the authors don't include any data on the speed the elephants were walking at other than 'normal' being used to classify it - if they didn't measure the speed then this is a serious flaw without speed information we can't assess the data in terms of the influence of the loads changes in speed will change gait so unless the speeds are the same with and without loads you can't answer the question posed if these data can be added then the manuscript can be re written 

Answer: We agree that this study would be stronger if we could measure the speed of the individual elephants; however, that was not possible in this study.

Therefore, we recognize this limitation in the Conclusions section (Lone 223-224),  

“We did not measure walking speeds to determine if that was affected by increased body mass loads, but based on video recordings, mahouts were not pushing the elephants harder when carrying extra weight.”.

Reviewer 3 Report

An elephant ride is a popular tourist activity, especially in Thailand, Cambodia, Myanmar and other parts of South-east Asia. The welfare of elephants used in tourism has been a topic of intense debate in recent years. The findings of this research can ease these debates of how elephants should be appropriately used in tourism, by endorsing that healthy elephants can comfortably carry  15% of its body weight without causing any discomfort or pressure to its musculoskeletal system and without significantly affect gait characteristics at the short distance. I enjoy reading the manuscript and am honored to be invited to review it . My (minor) suggestions to the authors are as follows:

The title of the manuscript did not properly reflect the subject of the paper. In sports science or in sports medicine “Kinematic analysis represents the position, motion, and trajectories of interest points of the subject to describe the locomotion pattern where mass and force are not considered” (details in Akhtaruzzaman, et. al (2016); Journal of Mechanics in Medicine and Biology 16(07): 1630003). “Gait” refers the pattern of movement of the limbs of animals (and humans) during locomotion over a solid substrate or on flat surface (Dicharry (2010); Clin Sports Med 29(3): 347-364). Gait is also referred to as foot fall pattern (Inuzuka 1996;Mammal Study 21(1): 43-57) and stride distance (distance between two consecutive ground contacts by the same limb) are always included in the assessment of gait kinematics. Although the manuscript is titled  “How does weight carriage impact gait kinematics in Asian elephants used for riding?”, the research related to foot fall pattern or stride distance is not included. I suggest to reword the title as “Impact of weight carriage on joint kinematics in Asian elephants used for riding on flat terrain”

Eight healthy adult elephants were used in the study. Table (1) showed demographic data. If the height data of these elephants are included in this study, it will help explain the variation of maximal flexion and extension angles and it can identify the position of the centre of gravity. Centre of gravity of an animal is high in tall animals.  The lower the centre of gravity  is, the more stable and easy for the animal to move and to control its stance. The higher it is the more likely the object is to tilt and in animals its more energy demanding to control the whole body mass to stable on its limbs. It will be more beneficial for the study if height data is utilized as a controlled variable.

Author Response

Reviewer#3

Comments and Suggestions for Authors

  1. An elephant ride is a popular tourist activity, especially in Thailand, Cambodia, Myanmar and other parts of South-east Asia. The welfare of elephants used in tourism has been a topic of intense debate in recent years. The findings of this research can ease these debates of how elephants should be appropriately used in tourism, by endorsing that healthy elephants can comfortably carry  15% of its body weight without causing any discomfort or pressure to its musculoskeletal system and without significantly affect gait characteristics at the short distance. I enjoy reading the manuscript and am honored to be invited to review it . My (minor) suggestions to the authors are as follows:

The title of the manuscript did not properly reflect the subject of the paper. In sports science or in sports medicine “Kinematic analysis represents the position, motion, and trajectories of interest points of the subject to describe the locomotion pattern where mass and force are not considered” (details in Akhtaruzzaman, et. al (2016); Journal of Mechanics in Medicine and Biology 16(07): 1630003). “Gait” refers the pattern of movement of the limbs of animals (and humans) during locomotion over a solid substrate or on flat surface (Dicharry (2010); Clin Sports Med 29(3): 347-364). Gait is also referred to as foot fall pattern (Inuzuka 1996;Mammal Study 21(1): 43-57) and stride distance (distance between two consecutive ground contacts by the same limb) are always included in the assessment of gait kinematics. Although the manuscript is titled  “How does weight carriage impact gait kinematics in Asian elephants used for riding?”, the research related to foot fall pattern or stride distance is not included.

I suggest to reword the title as “Impact of weight carriage on joint kinematics in Asian elephants used for riding on flat terrain”

Answer: Thank you very much for your comment. We have changed the title to Impact of weight carriage on joint kinematics in Asian elephants used for riding, but we did not say “on flat terrain” because as written it makes it sound like the elephants were routinely use for riding only on flat terrain, which is not the case. However, we do acknowledge that for this study, the elephants were walking on flat terrain in several places:

E.g., “Elephants walked at a steady speed in a straight line for 40 m on a flat grass field at their normal pace. “

“Therefore, it could be implied that for elephants, carrying 15% of body mass loads does not impact musculoskeletal function, at least during short walking bouts on flat terrain.”

“Our results suggest that loading an elephant’s back to the degree used in this study does not significantly affect gait characteristics over a short distance and on flat terrain”.

“And that Further investigations also are needed using larger numbers of elephants to explore longer-term impacts of saddle-riding on elephant structural and locomotor health; for instance, when walking over varying terrain and longer distances as is typically the case in Thailand.”

  1. Eight healthy adult elephants were used in the study. Table (1) showed demographic data. If the height data of these elephants are included in this study, it will help explain the variation of maximal flexion and extension angles and it can identify the position of the centre of gravity. Centre of gravity of an animal is high in tall animals.  The lower the centre of gravity  is, the more stable and easy for the animal to move and to control its stance. The higher it is the more likely the object is to tilt and in animals its more energy demanding to control the whole body mass to stable on its limbs. It will be more beneficial for the study if height data is utilized as a controlled variable.

Answer: Thank you very much for your comment. The height data were added in Table 1.

Reviewer 4 Report

The topic is important and the data are useful.
The introduction of the paper is generally good. Some key references are omitted- in particular, Asian elephant locomotor biomechanics have already been studied with 3D motion capture and IMUS, and force platforms and pressure pads-- the Genin and Ren 2 papers come up later but should come in here (~ref 14-18 in current text), plus examples such as:

Ren, L., Miller, C., Lair, R., Hutchinson, J.R. 2010. Integration of biomechanical compliance, leverage, and power in elephant limbs. Proceedings of the National Academy of Sciences USA 107:7078-7082. 

Ren, L. and Hutchinson, J.R. 2008. The three-dimensional locomotor dynamics of African (Loxodonta africana) and Asian (Elephas maximus) elephants reveal a smooth gait transition at moderate speed. Journal of the Royal Society- Interface 5:195–211. doi: 10.1098/​rsif.2007.1095

BW is not a weight but a mass (kg); please amend this throughout the MS. How was mass known? How precisely?

Table 1 should say Sex not Gender; as far as humans know, elephants do not have gender identities.

"A video camera also captured walking"-- give camera info; resolution and Hz.

Please explain in more detail, ideally with a figure, how the angles were computed. A 3D coordinate system showing the world, body and segment CSs should be made explicit. Better explanation for readers that do not know these IMUs is needed to explain how one goes from the raw data to the angles and ROMs. As written, it is not possible to precisely determine what each angle is relative to in the global/local CSs (e.g. in an elephant's anatomical context). Without a sensor at the pelvis how can motions between occiput and pelvis be deciphered, to understand hindlimb motions in a body CS? This seems a key problem. This limitation at least should be acknowledged.

If nonparametric tests were used wouldn't it be more appropriate to report median data not mean? Is adjustment for multiple comparisons needed?

The whole-forelimb/hindlimb ROMs (are these for the humerus/femur segment or both sensors per limb?) seem very small at ~5-7 degrees and SDs that bring the means ("confidence intervals") close to/overlapping with zero ROM. This is a concern that the data are faulty.

Except as noted above, the Discussion is adequate.

Please clarify what is meant here-- "Finally, the IMUs and software used were not developed for elephants, but rather were adapted from human studies evaluating bipedal motion, so additional refinements may be necessary."-- what is the study suggesting here? What might need refinement and why? Readers are left to wonder.

A limitation not noted is that kinetics might be more important for elephants than kinematics; e.g. forces and moments. Kinematics may not change but surely the kinetics do as the masses do (F= m*a). Perhaps even IMU data could address this; other studies of locomotion have done so.

It would be ideal if the raw+processed dataset of the study was made openly available to the community for maximal transparency, repeatability and re-use.

Author Response

Reviewer#4

  1. The topic is important and the data are useful.
    The introduction of the paper is generally good. Some key references are omitted- in particular, Asian elephant locomotor biomechanics have already been studied with 3D motion capture and IMUS, and force platforms and pressure pads-- the Genin and Ren 2 papers come up later but should come in here (~ref 14-18 in current text), plus examples such as:

Ren, L., Miller, C., Lair, R., Hutchinson, J.R. 2010. Integration of biomechanical compliance, leverage, and power in elephant limbs. Proceedings of the National Academy of Sciences USA 107:7078-7082. 

Ren, L. and Hutchinson, J.R. 2008. The three-dimensional locomotor dynamics of African (Loxodonta africana) and Asian (Elephas maximus) elephants reveal a smooth gait transition at moderate speed. Journal of the Royal Society- Interface 5:195–211. doi: 10.1098/​rsif.2007.1095

Answer: Thank you very much for your comment. These two articles have been added and cited.

Line 80-85

Ren and Hutchinson, who developed the IMU system, used it to quantify the locomotor pattern at normal and faster speeds in Asian and African elephants, measuring the center of mass motion and torso rotation. They revealed that kinematic and kinetic patterns were similar, and that the metabolic cost of walking was minimal at a normal speed, but it changed gradually with increasing speed of locomotion from inverted pendulum at slow speeds to a more bouncing gait at increasing speeds. Elephants retained a lateral sequence footfall pattern, which helped modulate energy output [20,21].

Line 131-132

Methodology was similar to the IMU method developed by Ren and Hutchinson [21].

Line 199-202

Ren et al. [20] used an IMU-based method to quantify kinematic and kinetic mechanisms and showed elephants use an effective energy-saving locomotion style by changing the center of mass pattern. The limbs’ effective mechanical advantage is increased by straightening the limbs, thus reducing ground reaction force moment arms.

With additional references

  1. Ren L, Miller CE, Lair R, Hutchinson JR. Integration of biomechanical compliance, leverage, and power in elephant limbs. Proc. Natl. Acad. Sci. U. S. A. 2010 April;13(107):7078–82.
  2. Ren L, Hutchinson JR. The three-dimensional locomotor dynamics of African (Loxodonta africana) and Asian (Elephas maximus) elephants reveal a smooth gait transition at moderate speed. J. R. Soc. Interface 2008 Feb;5(19):195-211.
  3. BW is not a weight but a mass (kg); please amend this throughout the MS. How was mass known? How precisely?

Answer: The word “body mass” has now been used throughout the manuscript.

  1. Table 1 should say Sex not Gender; as far as humans know, elephants do not have gender identities.

Answer: Yes, you are of course correct and we apologize for the oversight. “Sex” is now used instead of “Gender” in Table 1.

  1. "A video camera also captured walking"-- give camera info; resolution and Hz.

Answer: Thank you very much for your comment. The camera information has been added.

Line 118-119

A video camera (Logitech C925-e Webcam, Lausanne, Switzerland; resolution 640*480 pixels, 30 Hz frame rate) also captured walking activity

  1. Please explain in more detail, ideally with a figure, how the angles were computed. A 3D coordinate system showing the world, body and segment CSs should be made explicit. Better explanation for readers that do not know these IMUs is needed to explain how one goes from the raw data to the angles and ROMs. As written, it is not possible to precisely determine what each angle is relative to in the global/local CSs (e.g. in an elephant's anatomical context). Without a sensor at the pelvis how can motions between occiput and pelvis be deciphered, to understand hindlimb motions in a body CS? This seems a key problem. This limitation at least should be acknowledged.

Answer: The IMU package protocol automatically calculates the angles from the raw data signal. (https://www.stt-systems.com/motion-analysis/inertial-motion-capture/isen/#the-imu). We contacted the company for more details, but they have not responded. We also agree that a sensor at the pelvis would have aided in a more accurate calculation of hindlimb motion. We have acknowledged this in the limitations in the Conclusion as follows:

Line 219: We also did not have a reference sensor at the pelvis to more accurately measure hindlimb angles.

In Table 2, we have now used Proximal Flexion, Proximal Extension, Distal Flexion, Distal Extension instead of Shoulder/Hip Flex, Shoulder/Hip Ext, Elbow/Knee Flex, Elbow/Knee Ext.

In Table 3: “proximal angles” was used instead of shoulder/hip angles.

  1. If nonparametric tests were used wouldn't it be more appropriate to report median data not mean? Is adjustment for multiple comparisons needed?

Answer: We now report data as a Median and standard error (SE) in Tables 2 and 3.  Our study aims to compare within subject for two conditions, between non-weight and with 15% carrying weight, so adjusting for multiple comparisons is not needed.

  1. The whole-forelimb/hindlimb ROMs (are these for the humerus/femur segment or both sensors per limb?) seem very small at ~5-7 degrees and SDs that bring the means ("confidence intervals") close to/overlapping with zero ROM. This is a concern that the data are faulty.

Answer: The angles from the accelerometer rely on the axis in relation to gravity, if we did not set the same zero or starting point, data might be differences and reflected to a small amount of overall average range of motion (ROM). To reduce uncleared data, we deleted the ROM data and Table 4 and report only angles.

  1. Except as noted above, the Conclusion is adequate.

Answer: Thank you very much for your comment

  1. Please clarify what is meant here-- "Finally, the IMUs and software used were not developed for elephants, but rather were adapted from human studies evaluating, so additional refinements may be necessary."-- what is the study suggesting here? What might need refinement and why? Readers are left to wonder.

Answer: We agree this sentence was confusing. We just meant that the method had been used in other species and so we have deleted this sentence.

  1. A limitation not noted is that kinetics might be more important for elephants than kinematics; e.g. forces and moments. Kinematics may not change but surely the kinetics do as the masses do (F= m*a). Perhaps even IMU data could address this; other studies of locomotion have done so. Can Siriphan address this? I must admit I googled both terms to determine what was the right one, but it was confusing to me. Sorry if I misinterpreted their uses, so please correct.

Answer: Thank you very much for your comment. “Kinetics” is more reliable and be able to detect the mechanic changes better than “kinematics”. But kinetics analysis normally needs special equipment like force plate form; therefore, kinematic is specific and more appropriate in field work, as our study.

  1. It would be ideal if the raw+processed dataset of the study was made openly available to the community for maximal transparency, repeatability and re-use.

Answer: Thank you very much for your comment. The processed data set was added.